# Moderate Altitude Residence Reduces Male Colorectal and Female Breast Cancer Mortality More Than Incidence: Therapeutic Implications?

**DOI:** 10.3390/cancers13174420

**Published:** 2021-09-01

**Authors:** Johannes Burtscher, Grégoire P. Millet, Kathrin Renner-Sattler, Jeannette Klimont, Monika Hackl, Martin Burtscher

**Affiliations:** 1Department of Biomedical Sciences, University of Lausanne, CH-1015 Lausanne, Switzerland; Johannes.Burtscher@unil.ch (J.B.); Gregoire.Millet@unil.ch (G.P.M.); 2Institute of Sport Sciences, University of Lausanne, CH-1015 Lausanne, Switzerland; 3Internal Medicine III, University Hospital Regensburg, D-93059 Regensburg, Germany; Kathrin.Renner-Sattler@klinik.uni-regensburg.de; 4Unit Demography and Health, Directorate Social Statistics, Statistics Austria, 1110 Vienna, Austria; Jeannette.Klimont@statistik.gv.at; 5Austrian National Cancer Registry, Directorate Social Statistics, Statistics Austria, 1110 Vienna, Austria; Monika.Hackl@statistik.gv.at; 6Department of Sport Science, University of Innsbruck, A-6020 Innsbruck, Austria

**Keywords:** altitude, climate, lifestyle, colorectal, breast, disease

## Abstract

**Simple Summary:**

Living at moderate altitudes has been reported to be associated with health benefits, including reduced mortality from male colorectal and female breast cancer. The aim of this study was to evaluate altitude-dependent incidence rates and to compare them with mortality rates of those cancers. We further explored whether altitude-associated differences in lifestyle behaviour exist. Analyses including all incidence cases and deaths over a 10-year observation period of an Alpine country (Austria) revealed that the age-standardized incidence and mortality rates of male colorectal cancer decreased by 24.0% and 44.2%, and that of female breast cancer by 6.5% and 26.2%, from the lowest (<251 m) to the highest (1000–2000) altitude level. The population-based survey indicated higher physical activity levels and lower body mass index for both sexes living at a moderate altitude compared to those living below 251 m. These observations may, in certain cases, support decision making when changing residence.

**Abstract:**

Background: Living at moderate altitude may be associated with health benefits, including reduced mortality from male colorectal and female breast cancer. We aimed to determine altitude-dependent incidence and mortality rates of those cancers and put them in the context of altitude-associated lifestyle differences. Methods: Incidence cases and deaths of male colorectal cancer (*n* = 17,712 and 7462) and female breast cancer (*n* = 33,803 and 9147) from altitude categories between 250 to about 2000 m were extracted from official Austrian registries across 10 years (2008–2017). Altitude-associated differences in health determinants were derived from the Austrian Health Interview Survey (2014). Results: The age-standardized incidence and mortality rates of male colorectal cancer decreased by 24.0% and 44.2%, and that of female breast cancer by 6.5% and 26.2%, respectively, from the lowest to the highest altitude level. Higher physical activity levels and lower body mass index for both sexes living at higher altitudes were found. Conclusions: Living at a moderate altitude was associated with a reduced incidence and (more pronounced) mortality from colorectal and breast cancer. Our results suggest a complex interaction between specific climate conditions and lifestyle behaviours. These observations may, in certain cases, support decision making when changing residence.

## 1. Introduction

Besides genetic and lifestyle factors, environmental (climate) conditions are relevant modulators of mortality and associated life expectancy [1,2,3,4,5]. Environmental conditions, i.e., barometric pressure, partial pressure of oxygen, temperature, ultra-violet (UV) radiation, and air pollution, change characteristically with varying altitudes [6]. In contrast, at high altitudes, i.e., >2500 m, and in particular, in high-altitude populations, such as Tibetans or Andeans, genetic adaptations to hypobaric hypoxia seem to play a major role in the modulation of mortality [7], interactions between all those environmental conditions and altered lifestyle factors (e.g., diet and physical activity) may, more importantly, impact on mortality and associated longevity at moderate altitudes, i.e., below 2500 m [8,9]. We previously demonstrated that age-standardized mortality rates (ASR-M) from male colorectal cancer and female breast cancer decreased almost linearly from low (<251 m) to higher (1001 to about 2000 m) altitude [8] by 45% and 38%, respectively. In that study, information was available neither on the altitude-related cancer incidence nor on the potential variation of lifestyle factors in the respective general population. Addressing this knowledge gap, however, may be crucial to understand moderate-altitude effects on disease development and mortality from those cancers and ultimately for the design of preventive/therapeutic strategies. A recent meta-analysis demonstrated more pronounced benefits of favourable combined lifestyle factors on cancer mortality than onincidence [10]. Therefore, we hypothesized that living at moderate altitude affects cancer incidence and mortality differently. To investigate this possibility, we analysed altitude-dependent incidence rates of male colorectal and female breast cancer (as done in a previous study, in which we reported data from an earlier observation period) [8] and compared them to mortality rates of an Alpine country (Austria) where altitudes of residence range from below 200 m up to about 2000 m. Moreover, to explore whether altitude-associated differences in lifestyle behaviour exist, a representative Austrian population-based health survey was interrogated.

## 2. Materials and Methods

### 2.1. Incidence and Mortality Data

Altitude-dependent incidence and mortality data from male colorectal and female breast cancer from 2008 until 2017 were extracted from the Austrian National Cancer Registry and the Austrian Causes of Death Statistics (both located at Statistics Austria). Data were compared for the following altitude categories: <251 m, 250–500 m, 501–750 m, 751–1000 m, >1000 m (1001 to about 2000 m). As in our previous paper [8], only communities with a population below 20,000 were included to avoid important confounding from migration.

The total numbers of the incidence cases and deaths for male colorectal cancer (ICD-10: C18-C21) were 17,712 and 7462 and for female breast cancer (ICD-10: C50) 33,803 and 9147, respectively. Age-standardized incidence (ASR-I) and mortality (ASR-M) rates per 100,000 population and 95% confidence intervals (CI), based on the assumption that the data follow a Poisson distribution, are reported (provided by Statistics Austria). Differences between mortality rates of our previous (years of diagnosis: 2003 to 2012) and the present study (years of diagnosis: 2008 to 2017) are due to the use of different standard populations; in contrast to the previous study, in which the calculations were based on the WHO world standard population, here, the European standard population was used for this purpose [11].

### 2.2. Health Interview Survey Results

In order to obtain information on potential altitude-associated differences in health determinants, such as body mass index, physical activity, diet, smoking and alcohol consumption, the report of the Austrian Health Interview Survey (ATHIS) 2014 [12] was consulted. For each survey respondent, the respective altitude of the place of residence has been added through data merging via the municipality code. ATHIS is a nationally representative study of persons aged over 15 years living in private Austrian households. Only data of individuals older than 19 years are included in the analyses of the present report.

Altitude-dependent data are presented descriptively for both sexes, and the indicated statistical differences (unpaired t-tests, Mann–Whitney U Test or chi-squared tests) refer to comparisons between values recorded at the lowest and highest altitude zones. Statistical analyses were performed by IBM SPSS version 26.0 (IBM SPSS Statistics for Windows, Chicago, IL, USA). *p*-values (2-sided) below 0.05 were considered statistically significant.

## 3. Results

### 3.1. Incidence and Mortality Data

The ASR-I and ASR-M (95% confidence intervals) of male colorectal cancer decreased by 17.7% and 35.5%, respectively, from the lowest to the highest altitude level. The decline of these indicators for female breast cancer was 6.8% and 17.4% (Table 1).

ASR-I and ASR-M of male colorectal cancer and ASR-M of female breast cancer decreased linearly from low to higher altitudes, whereas the ASR-I of female breast cancer remained essentially unchanged up to 1000 m and then slightly declined. Altitude-dependent different percentage changes (compared to <251 m) between incidence and mortality are shown in Figure 1A for male colorectal cancer and in Figure 1B for female breast cancer.

### 3.2. Health Interview Survey Results

Next, we aimed to test the hypothesis that average population behaviours with the potential to influence the risk of cancer incidence and mortality are also altered at different altitudes of residence. To this end, we analysed data from a highly representative health survey on the Austrian population conducted during our observation period in 2014. Items with relevance to cancer morbidity and mortality were selected and analysed for the altitude levels applied in the previously presented analyses (Table 2).

Dietary habits were significantly influenced by the altitude of residence. While daily vegetable and fruit intake were particularly low in the lowest altitude category (<251 m) for both men and women, daily fruit intake in men and vegetable intake in women increased continuously with increasing altitude of residence.

A strong altitude dependence was also observed for alcohol intake and smoking habits that were affected by altitude in opposite directions in men and women. Whereas daily smoking and at least bi-weekly alcohol consumption were positively correlated with increasing altitude for men, these behaviours were reduced in women at higher altitudes as compared with the lowest altitude category.

Physical activity levels were significantly higher at altitudes of >1000 m as compared to <251 m for both men and women, and this effect was stronger for men, who, on average, were about 41% more physically active if residing at >1000 m as compared to their lowland counterparts. Men and women living higher than 1000 m also had lower body mass indexes on average.

Altitude-dependent differences in the overall quality of life were negligible.

Of interest, the percentage of individuals who had never made use of relevant preventive medical check-ups was lower at higher altitudes; no significance was observed for colonoscopies (men), but significantly more women at >1000 m (as compared to the <251 m category) never had mammography.

## 4. Discussion

The main findings of the present analyses are (1) a stronger benefit of moderate altitude residence on cancer mortality rates than on incidence rates, and (2) both rates decreased more steeply with increasing altitude for male colorectal cancer as compared to female breast cancer, and (3) there are distinct altitude-specific differences in lifestyle behaviours of the general population, likely contributing to the observed “altitude effects” on cancer incidence and mortality.

These findings are in accordance with those derived from a recent review, which demonstrated that exposure to high altitude reduces both incidence and mortality from several types of cancer [13]. However, sufficient consideration of all potential confounders, such as ethnicity, industrialization, urbanization, socioeconomic and sociocultural status as well as lifestyle behaviours, is extremely difficult, as exemplified by contrasting results of two large-scale epidemiological studies. One performed in China reported the lowest mortality rates for certain types of cancers, i.e., lung, colon and rectum in both sexes, and breast cancer in females, in the highest living population (Tibetans, average altitude of about 4000 m) [14], while another found elevated cancer prevalence and mortality (including colorectal and breast cancers) in the Ecuadorian population also living at high altitude (2000 to > 4600 m) [15]. These discrepancies may be also associated with the fact that populations native to the Tibetan and Andean Plateaus have adapted (natural selection process) differently to the environmental stress of severe lifelong high-altitude hypoxia [16]. As most studies evaluated effects of altitudes above 2500 m, the reduced oxygen partial pressure was suggested primarily to account for the observed effects on cancer incidence and mortality [13]. However, this may not be true for the moderate altitudes of our study population, ranging between 200 and about 2000 m. The only mild hypobaric hypoxia in the higher regions is accompanied by lower ambient temperature and air pollution and higher solar radiation. Apart from these climate conditions, the altitude-related differences in lifestyle behaviour may also contribute to the observed benefits of moderate altitude residence on cancer incidence and mortality rates.

### 4.1. Climate Conditions of Moderate-Altitude Potentially Affecting Cancer Incidence and Mortality Rates

Although lifestyle factors, such as diet and physical activity/exercise modulate cancer risk (see below), environmental factors (e.g., hypobaric hypoxia, level of vitamin D, pollution) of higher altitudes have been suggested to play a major role for discrepancies of several diseases, including lower incidence and mortality of cardiovascular diseases [17], diabetes mellitus [18], obesity [19], metabolic syndrome [20,21], as well as cancer mortality at different altitudes [8,22]. Altitude affects health by changes in and complex interactions between the specific terrain (physical challenge), lifestyle, socioeconomic and in particular environmental (climatic) conditions. Increasing levels of hypoxia, cold (ambient temperature is decreased by about 6.5 °C per 1000 m gain in altitude [23]) and solar radiation (if the sky is clear, global irradiation is increased by about 8% per 1000 m gain in altitude [24]) with altitude are closely interrelated and cannot be easily disentangled when interpreting findings on cancer frequencies from epidemiological studies.

With increasing altitude, the barometric pressure and the related inspiratory, alveolar and arterial oxygen partial pressures decrease. Although there is only a slight decline in oxygen saturation at moderate altitude based on the oxygen-haemoglobin dissociation curve, the reduction may be more pronounced during conditions, such as physical activity or sleep [25,26]. Thus, hypoxia-related activation of the hypoxia-inducible factor (HIF) pathway may occur even at moderate altitude. In this case, the expression of dozens of genes is modulated to support the maintenance of tissue oxygen supply [27], likely differently affecting the development and progression of cancer [6,13,28].

People living at moderate or higher altitudes are usually well protected from unfavourable health effects of cold, e.g., by heated homes and appropriate clothing, but some exposure during work, exercise, etc., is unavoidable and may cause adaptations/habituations. Such adaptations include, e.g., blunted sympathetic stress responses and improved stress tolerance but also increased basal metabolic rates, likely counteracting obesity [29,30] and the associated cancer risks [31]. In addition, recent studies indicate that solar UV radiation, which increases with altitude, favourably impacts the human organism, e.g., by improving cardiovascular health and attenuating cancer development, likely mediated by the role of UV in vitamin D synthesis [32,33]. Hypobaric hypoxia of moderate altitude may indirectly modify cancer incidence and mortality by reducing cardiovascular risk factors [8] and directly by acquired tolerance to hypoxia (e.g., hypoxic preconditioning), as has been argued using the hypoxia-tolerant naked mole rat [34]. However, how hypoxia per se contributes to the observed differences between cancer types requires elucidation.

Levels of hypoxia [35] and UV radiation [36] were shown to be positively and ambient temperatures [37] negatively correlated with the risk of certain cancers. Our results indicate that potential adverse climate influences do not necessarily determine male colorectal and female breast cancer incidence and mortality risks, at least at moderate altitudes. It is thus possible that at such altitudes, non-climate factors become the dominant modulators of the risk to develop or die from these cancers.

### 4.2. Non-Climate Factors Potentially Modifying Cancer Incidence and Mortality Rates

Lifestyle behaviour, such as diet and physical activity, play important roles in the development of colorectal cancers. Several foods and nutrients, e.g., fibres, milk, calcium, whole grains, vitamin D, fruits and vegetables, have been linked to a reduced, and, e.g., red meat and processed meat, to an elevated risk for colorectal cancer [38]. Moreover, the risk of suffering from any type or from advanced colorectal cancer was demonstrated to be 23% or 27%, respectively, lower in people with the highest level compared to those with the lowest level of physical activity [39]. Obesity and smoking were positively and the use of non-steroidal antirheumatics negatively associated with the risk of colorectal cancer [40]. Cigarette smoking and alcohol consumption seem to increase the risk of colorectal cancer in an additive manner, indicating that both either share a common etiologic pathway in carcinogenesis [41] or increase cancer risk independently.

Long-term physical activity may beneficially impact both the risk of colorectal and breast cancer to a similar extent (20% to 30% risk reduction) [31]. In contrast to colorectal cancer, the risk of breast cancer seems associated to a lesser extent with the intake of meat or dairy products [42]. While vitamin D blood levels were slightly negatively correlated with breast cancer risk (an increase of blood vitamin D levels of 5 nmol/L reduced breast cancer risk by 6%), supplementation of vitamin D was not significantly associated with breast cancer risk in a recent meta-analysis [43]. Smoking was associated with an about 10% risk increase derived from prospective studies [44]. A small effect of alcohol but no independent effect of smoking on the risk of female breast cancer development was reported in another meta-analysis [45].

A positive association between higher levels of dairy products, fruit and vegetable intake as well as physical activity with the increasing residence of altitude and/or rurality seems plausible [46]. Thus, the recorded altitude-dependent differences in diet, smoking, alcohol drinking and physical activity in an Alpine population could contribute to the explanation of demonstrated incidence and mortality rates of colorectal and breast cancer well. These new observations moderate the assumption that beneficial effects of moderate altitude are almost exclusively attributed to environmental conditions [8,9]. Here, we demonstrate that this viewpoint is too simplistic since both incidence and mortality are likely influenced by several interrelated factors. In the studied population, the altitude of residence was significantly associated—but in a contradictory way between men and women—with dietary, smoking and alcohol consumption habits, all potential modulators of cancer. However, both men and women living higher on average reported markedly higher physical activity levels and reduced body mass index. In light of the highly protective effect of physical activity on cancer prevention [47] and mortality [48] in general, these results may partly explain the reduced incidence and mortality rates of male colorectal cancer and reduced mortality rates of female breast cancer with increasing altitude. Moreover, this later result suggests that physical activity may be the most important lifestyle factor.

### 4.3. Factors Potentially Explaining Differences between Cancer Incidence and Mortality Rates

Our data indicate that living at moderate altitude affects ASR-M more strongly than ASR-I. Despite the observation of major altitude dependences of lifestyle factors that contribute to cancer risk, climate conditions—likely including mild hypoxia—should still be considered important contributing factors potentially explaining the results of the present study, in particular in association with physical activity.

A potential beneficial role of environmental hypoxia at first seems to be contradictory to HIF-1 overexpression and associated tumour angiogenesis, metastasis, and invasion due to tumour-related hypoxia in several solid tumours, including colorectal and breast cancers [28]. Importantly, tumour-hypoxia results in dysfunctional vascularization and favours cell mobility and metastasis due to epithelial-to-mesenchymal transition [49], which differs from effects of physical activity and ambient hypoxia at altitude. Physical activity after diagnosis of colorectal or breast cancer can indeed reduce the risk of death by about 50% [50,51]. Although underlying mechanisms are discussed controversially, aerobic exercise may increase intra-tumoral vascularization resulting in “normalization” of the tissue microenvironment mediated by vascular endothelial growth factor (VEGF) [52]. In this animal study, histological analyses indicated higher intra-tumoral hypoxia levels (assessed by HIF-1 expression) in the group performing exercise versus controls. Exercise may induce hypoxia through the exercise-induced hypoxemia mechanism [53] and associated VEGF and angiogenesis, which is aggravated when exercising in hypoxia (likely even at moderate altitude [54]) [55]. Such normalization of tumour vascularization was suggested to reduce glycolysis, thereby inhibiting distant metastases [56]. Moreover, normal perfusion of the tumoral tissue may support the delivery and efficacy of systemic and regional (e.g., chemotherapy, radiotherapy) anticancer therapies [46]. Observed differences between cancer incidence and mortality may also be attributable to other climate and non-climate related factors. For instance, an updated meta-analysis confirmed the beneficial effects of adherence to a Mediterranean diet, i.e., the intake of fruits, vegetables and whole grains, especially on the mortality from colorectal cancer. Another meta-analysis revealed a clinically meaningful benefit of higher vitamin D levels on survival from colorectal cancer [57] and breast cancer as well [58], potentially associated with UV action at altitude on Vitamin D synthesis [33].

Finally, the demonstrated differences in preventive medical examinations are suggestive of potential misrepresentation of cancer incidence at higher altitudes of residence. However, the observed smaller risk reductions of incidence as compared to mortality at higher altitudes, on the contrary, could indicate an underestimation of cancer incidence and thus an even more pronounced real difference to mortality.

### 4.4. Limitations

Based on the analyses of the complete dataset on the Austrian incidence and mortality from colorectal cancer in men and breast cancer in women across 10 years, the novel findings of a pronounced divergence of incidence and mortality in these two types of cancer represent the main strengths of the present study. However, we have to acknowledge some limitations. The analyses were selectively performed on data of male colorectal and female breast cancers based on our previous findings [8]. The effects of moderate altitude residence on the incidence and mortality of other cancers remain to be elucidated, and the study has not been validated by considering colorectal cancer in women. The well-established sex-and age-related differences in incidence and mortality of colorectal cancer [59,60] will make such analyses particularly interesting. As discussed above, the epidemiological data, on which our analyses are based, are not suitable for the identification of causal explanations for the observed effects. As the results of the Austrian health survey indicate, the various altitude levels are associated with numerous differences in lifestyle, many of which potentially modulate cancer risk independently. Although we are aware of this problem, a statistical identification of explanatory variables is not feasible because data have been derived from two separate data sets (complete population data of mortality and incidence vs. representative snapshot of health survey). The confirmation of the reported associations of cancer incidence and mortality with different lifestyles at distinct altitude levels by future studies is therefore needed.

## 5. Conclusions

Living at moderate altitude is associated with a pronounced reduction of ASR-I and ASR-M of male colorectal and female breast cancer, with a steeper decline of ASR-M than ASR-I and with a steeper decline of both rates for male colorectal cancer. Besides altitude-related environmental conditions, including mild hypobaric hypoxia, results of the population-based health survey indicate that lifestyle behaviours, such as diet and physical activity, are likely contributing factors to the modulation of the ASR-I and ASR-M. These results suggest that moderate altitude residence reduces the risk to die from colorectal or breast cancer to a stronger degree than it protects from contracting it (although it seems to be protective also in that regard). This implies that prolonged exposure to altitude-related climate conditions but also the changes in lifestyle factors associated with different moderate-altitude categories could ameliorate the cancer outcome even after a cancer diagnosis. These observations may, in certain cases, support decision making when changing residence. For the successful implementation of interventional strategies, further studies are needed to disentangle independent and interaction effects of climate and non-climate factors at a moderate altitude, explaining the observed beneficial effects on cancer incidence and mortality.

## Figures and Tables

**Figure 1 cancers-13-04420-f001:**
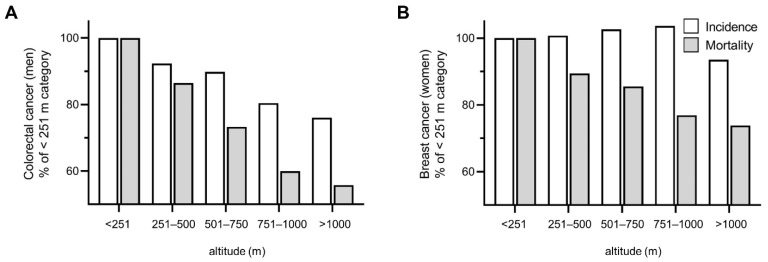
Age-standardized incidence and mortality rates of (**A**) colorectal cancer in men and (**B**) breast cancer in women, according to the altitude of residence. Percentages in relation to the lowest altitude category (<251 m) are presented.

**Table 1 cancers-13-04420-t001:** Age-standardized incidence and mortality rates for colorectal cancer in Austrian men and breast cancer in Austrian women in dependence of the altitude of residence.

	Colorectal Cancer, Men			Breast Cancer, Women		
Altitude (m)	Incidence	SE	95% CI	Incidence	SE	95% CI
<251	88.2	1.6	85.1;91.3	122.8	1.6	119.7;125.9
251–500	81.4	1.0	79.4;83.4	123.7	1.1	121.5;125.9
501–750	79.2	1.2	76.9;81.6	126.1	1.3	123.6;128.6
751–1000	70.9	1.9	67.2;74.6	127.4	2.3	122.9;131.9
>1000	67.0	3.1	60.9;73.1	115.3	3.7	108.1;122.6
	**Mortality**	**SE**	**95% CI**	**Mortality**	**SE**	**95% CI**
<251	45.0	1.2	42.7;47.4	35.9	0.9	34.1;37.7
251–500	38.9	0.7	37.5;40.3	32.1	0.5	31.1;33.1
501–750	33.0	0.8	31.4;34.6	30.7	0.7	29.3;32.1
751–1000	27.0	1.2	24.7;29.4	27.6	1.1	25.4;29.8
>1000	25.1	2.0	21.2;29.0	26.5	1.8	23.0;30.0

Incidence and mortality for men (colorectal cancer) and women (breast cancer) from Austrian communities with a population of not greater than 20,000 inhabitants are depicted according to the altitude of residence for the years of diagnosis 2008–2017. SE-standard error, CI-confidence interval.

**Table 2 cancers-13-04420-t002:** The selected results derived from the representative, population-based, health interview survey.

	Men					Women				
Altitude, m	<251	251–500	501–750	751–1000	>1000	<251	251–500	501–750	751–1000	>1000
Number	1361	2876	1752	520	204	1749	3614	2171	702	263
Body mass index, kg/m^2^	27 (4)	26 (4)	26 (4)	26 (4)	25 (4) *	25 (5)	25 (5)	255 (5)	25 (5)	24 (4) *
Smokers, daily, %	17	19	25	24	28 *	25	22	18	19	18 *
Alcohol consumption on ≥ 2 days/week, %	20	21	28	29	27 *	11	9	8	5	7 *
Fruit intake, daily, %	43	46	48	49	51 *	62	66	70	69	64
Vegetable intake, daily, %	33	41	42	43	39	47	60	60	61	57 *
Physical activity, min/week	195 (316)	199 (248)	222 (302)	232 (294)	275 (520) *	163 (222)	171 (222)	202 (260)	207 (239)	194 (286) *
QoL, score (1–5)	4.1 (0.8)	4.1 (0.7)	4.1 (0.7)	4.2 (0.7)	4.2 (0.7)	4.1 (0.8)	4.1 (0.8)	4.1 (0.8)	4.2 (0.7)	4.1 (0.7)
Medical examination	% of men, who never had a colonoscopy					% of women who never had a mammography				
Age >19 years	63	62	63	60	59	26	27	26	28	33 *
Age >50 years	44	41	39	35	37					

Data are presented as means (±SD) or frequencies (percentages). * *p* < 0.05 (>1000 m vs. <251 m). QoL: quality of life.

## Data Availability

www.statistik.at (accessed on 31 August 2021).

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
