# Peer review of "Moderate Altitude Residence Reduces Male Colorectal and Female Breast Cancer Mortality More Than Incidence: Therapeutic Implications?"

_cancers, 2021, doi:10.3390/cancers13174420_

Round 1

Reviewer 1 Report

Summary

Burtscher et al. studied the effects of altitude and cancer incidence and mortality (colorectal cancer in men and breast cancer in women) in a sample from the Austrian population at moderate altitude. They found that age-standardized incidence and mortality rates of colorectal cancer decreased by 17.7% and 35.5%, and that of breast cancer by 6.8% and 39 17.4%, respectively, from the lowest (<251m) to the highest altitude level (1000-2000). They also found that subjects living at higher altitudes have higher physical activity levels and lower body mass index. This paper adds data to their 2016 publication (Aging (Albany NY). 2016 Oct; 8(10): 2603–2604.) They have previously shown that male colorectal cancer female breast cancer decreased in higher altitudes. In this current publication, they have included the following data from the same population:

1) Altitude-related cancer incidence

2) Lifestyle factors

As a result of information about lifestyle factors, they revealed a correlation between altitude and lifestyle behavior. They report that climate conditions of moderate altitude did not explain cancer incidence and mortality rates. However, they found that physical activity correlated with reduced colorectal and breast cancer incidence in their sample population.

Broad comments:

Combining cancer incidence and mortality with modifiable lifestyle data from a well-defined population revealed a stronger correlation with mortality than incidence. As the authors mentioned, the results should be interpreted with caution because of the cross-sectional source of the lifestyle data.

Physical activity and diet are modifiable risk factors and are helpful for public health interventions. These interventions have been applied irrespective of altitudes to improve cancer outcomes such as recurrence and survival (References 50 & 51). Although many reports evaluated the relationship between altitude and health outcomes, Burtscher et al.'s data add to the literature and would be helpful for behavioral modification interventions to reduce mortality, morbidity, and potential incidence of cancer.

A follow-up of the survey respondents and cancer registry would further clarify this correlation.

Specific comments:

  • Describe how the altitude data was extracted from the health survey?
  • Add more information about the statistical analysis method, such as software?
  • Line 105: is the p-value one-sided or two?
  • Table 2:
    • no need for decimal points as it affects the readability of the data
    • Rewrite the following phrases:
  1. "% of men never having made use of a colonoscopy": e.g., % of men never had a colonoscopy
  2. "% of women never having made use of a mammography"
  3. "p<0.05 for values"

Author Response

Dear Reviewer,

first of all, we would like to thank for the really helpful and constructive comments that helped us to improve our manuscript!

We tried to respond adequately to all of your points (please see below) and revised the manuscript accordingly. We provide a revised version with changes in the tracked modus and a clean one.

Reviewer 1

Summary

Burtscher et al. studied the effects of altitude and cancer incidence and mortality (colorectal cancer in men and breast cancer in women) in a sample from the Austrian population at moderate altitude. They found that age-standardized incidence and mortality rates of colorectal cancer decreased by 17.7% and 35.5%, and that of breast cancer by 6.8% and 39 17.4%, respectively, from the lowest (<251m) to the highest altitude level (1000-2000). They also found that subjects living at higher altitudes have higher physical activity levels and lower body mass index. This paper adds data to their 2016 publication (Aging (Albany NY). 2016 Oct; 8(10): 2603–2604.) They have previously shown that male colorectal cancer female breast cancer decreased in higher altitudes. In this current publication, they have included the following data from the same population:

1) Altitude-related cancer incidence

2) Lifestyle factors

As a result of information about lifestyle factors, they revealed a correlation between altitude and lifestyle behavior. They report that climate conditions of moderate altitude did not explain cancer incidence and mortality rates. However, they found that physical activity correlated with reduced colorectal and breast cancer incidence in their sample population.

Broad comments:

Combining cancer incidence and mortality with modifiable lifestyle data from a well-defined population revealed a stronger correlation with mortality than incidence. As the authors mentioned, the results should be interpreted with caution because of the cross-sectional source of the lifestyle data.

Physical activity and diet are modifiable risk factors and are helpful for public health interventions. These interventions have been applied irrespective of altitudes to improve cancer outcomes such as recurrence and survival (References 50 & 51). Although many reports evaluated the relationship between altitude and health outcomes, Burtscher et al.'s data add to the literature and would be helpful for behavioral modification interventions to reduce mortality, morbidity, and potential incidence of cancer.

A follow-up of the survey respondents and cancer registry would further clarify this correlation.

Re: We thank the reviewer for highlighting the novelty and potential usefulness of or study for public health interventions and yes, we actually will consider a follow-up study.

Specific comments:

 Describe how the altitude data was extracted from the health survey?

Re: For each survey respondent the respective altitude of the place of residence has been recorded. This information is now provided in the manuscript.  

  • Add more information about the statistical analysis method, such as software?

Re: We included as follows: Statistical analyses were performed by IBM SPSS version 26.0 (IBM SPSS Statistics for Windows, Chicago, IL, USA).

  • Line 105: is the p-value one-sided or two?

Re: Thank you for this point. All p-values reported are 2-sided. This information is now given in the statistical methods.

  • Table 2:
    • no need for decimal points as it affects the readability of the data
    • Rewrite the following phrases:
  1. "% of men never having made use of a colonoscopy": e.g., % of men never had a colonoscopy
  2. "% of women never having made use of a mammography"
  3. "p<0.05 for values"

Re: We thank the reviewer for these suggestions and changed table 2 accordingly. We only report one decimal place for the QoL data due to the small differences between altitudes.

Table 2 now reads as follows:

Table 2. Selected results derived from the representative, population-based, health interview survey.

Men

Women

Altitude, m

<251

251-500

501-750

751-1000

>1000

<251

251-500

501-750

751-1000

>1000

Number

1361

2876

1752

520

204

1749

3614

2171

702

263

Body mass index, kg/m2

27 (4)

26 (4)

26 (4)

26 (4)

25 (4)*

25 (5)

25 (5)

25 (5)

25 (5)

24 (4)*

Smokers, daily, %

17

19

25

24

28*

25

22

18

19

18*

Alcohol consumption on ≥ 2 days/week, %

20

21

28

29

27*

11

9

8

5

7*

Fruit intake, daily, %

43

46

48

49

51*

62

66

70

69

64

Vegetable intake, daily, %

33

41

42

43

39

47

60

60

61

57*

Physical activity, min/week

195

(316)

199

(248)

222

(302)

232

(294)

275

(520)*

163

(222)

171

(222)

202

(260)

207

(239)

194

(286)*

QoL,

score (1-5)

4.1 (0.8)

4.1 (0.7)

4.1 (0.7)

4.2 (0.7)

4.2

(0.7)

4.1 (0.8)

4.1 (0.8)

4.1 (0.8)

4.2 (0.7)

4.1 (0.7)

Medical examination

% of men,

who never had a colonoscopy

% of women,

who never had a mammography

Age > 19 years

63

62

63

60

59

26

27

26

28

33*

Age > 50 years

44

41

39

35

37

Data are presented as means (±SD) or frequencies (percentages).

* p<0.05 (>1,000 m vs <251 m). QoL: quality of life.

Reviewer 2 Report

In this simple ecological study, the authors investigate the relationship between altitude of residence and incidence/mortality of CRC (only for men) and breast cancers. They found a linear gradient of decrease in the risk of mortality from these cancers with increasing altitude. On the other hand, they found a significant association between altitude and the prevalence of exposure to certain known risk factors for these diseases (obesity, inactivity ...). From these two separate analyses, they suggest that part of the effects of altitude could be mediated by these factors.

General comments: The content of the article is interesting and well written, but leaves the reader somewhat frustrated. The analyses carried out in the article are only a preamble to assess to what extent the differences in cancer incidence/mortality between different altitudes are due to non-climatic factors. As suggested in the discussion, risk factors such as obesity, alcohol or tobacco consumption are likely to be important mediators of the effects of altitude on mortality. The interest of the article would benefit from evaluating to what extent the effect of altitude is maintained after taking into account differences in life behaviour.

We agree that such an analysis could have limitations in a causal setting given the ecological nature of the measure of exposure to risk factors (prevalence). However, we would prefer that these limitations were given to qualify the results of a mediation analysis rather than to justify not carrying it out.

Minor comments:

The risk reduction given in the abstract is not consistent with the results given in Table 1. It appears that the authors used the category "251-500 m" as a reference for this calculation rather than the lowest altitude level as indicated in the text.

Table 1: please indicate in the legend that incidence and mortality were age-standardized.

The relationship between the incidence of breast cancer and altitude is essentially constant or at least without significant variations. The allusion to a "U" shape for this relation does not seem justified.

Please justify why the CRC was studied only for men.

Author Response

Dear Reviewer,

first of all, we would like to thank for the really helpful and constructive comments that helped us to improve our manuscript!

We tried to respond adequately to all of your points (please see below) and revised the manuscript accordingly. We provide a revised version with changes in the tracked modus and a clean one.

Reviewer 2

In this simple ecological study, the authors investigate the relationship between altitude of residence and incidence/mortality of CRC (only for men) and breast cancers. They found a linear gradient of decrease in the risk of mortality from these cancers with increasing altitude. On the other hand, they found a significant association between altitude and the prevalence of exposure to certain known risk factors for these diseases (obesity, inactivity ...). From these two separate analyses, they suggest that part of the effects of altitude could be mediated by these factors.

General comments: The content of the article is interesting and well written, but leaves the reader somewhat frustrated. The analyses carried out in the article are only a preamble to assess to what extent the differences in cancer incidence/mortality between different altitudes are due to non-climatic factors. As suggested in the discussion, risk factors such as obesity, alcohol or tobacco consumption are likely to be important mediators of the effects of altitude on mortality. The interest of the article would benefit from evaluating to what extent the effect of altitude is maintained after taking into account differences in life behaviour.

We agree that such an analysis could have limitations in a causal setting given the ecological nature of the measure of exposure to risk factors (prevalence). However, we would prefer that these limitations were given to qualify the results of a mediation analysis rather than to justify not carrying it out.

Re: We thank the reviewer for the interest in our study and fully agree that the lack of linking the risk factors with the epidemiology is a major limitation. However, these data have been derived from two separate data sets, rendering this analysis impossible. We hope that in the revised version the limitations are now clearer (see modified limitations section). Please find below the answers to the minor comments.

Minor comments:

The risk reduction given in the abstract is not consistent with the results given in Table 1. It appears that the authors used the category "251-500 m" as a reference for this calculation rather than the lowest altitude level as indicated in the text.

Re: Thank you for making us aware of this mistake: data from previous analyses had not been updated in the abstract here. This has now been corrected and reads as follows:

Analyses including all incidence cases and deaths over a 10-year observation period of an Alpine country (Austria) revealed that the age-standardized incidence and mortality rates of male colorectal cancer decreased by 24.0% and 44.2%, and that of female breast cancer by 6.5% and 26.2%, from the lowest (<251 m) to the highest (1,000 – 2,000) altitude level.)

Thank you very much for pointing out this error.

Table 1: please indicate in the legend that incidence and mortality were age-standardized.

Re: Has been done – also in the figure legend.

The relationship between the incidence of breast cancer and altitude is essentially constant or at least without significant variations. The allusion to a "U" shape for this relation does not seem justified.

Re: We agree with the reviewer and changed accordingly.

Please justify why the CRC was studied only for men.

Re: We thank the reviewer for this important point. The main reason to do so was to provide a direct comparison with data from an earlier observation period (reference 8), in which already altitude effects on those cancers were reported. We refer to that at the end of the intro section.

Thank you very much again!